# Assessment of Cardiovascular Function in Childhood Leukemia Survivors: The Role of the Right Heart

**DOI:** 10.3390/children9111731

**Published:** 2022-11-11

**Authors:** Paola Muggeo, Pietro Scicchitano, Vito Michele Rosario Muggeo, Chiara Novielli, Paola Giordano, Marco Matteo Ciccone, Maria Felicia Faienza, Nicola Santoro

**Affiliations:** 1Department of Pediatric Oncology and Hematology, University Hospital of Policlinico, 70124 Bari, Italy; 2Cardiovascular Diseases Section, Department of Emergency and Organ Transplantation (DETO), University of Bari “A. Moro”, 70124 Bari, Italy; 3Department of Economical, Business and Statistical Sciences, University of Palermo, 90128 Palermo, Italy; 4Department of Interdisciplinary Medicine, Pediatric Unit, University of Bari “A. Moro”, 70124 Bari, Italy; 5Department of Precision and Regenerative Medicine and Ionian Area, Pediatric Unit, University of Bari “A. Moro”, 70124 Bari, Italy

**Keywords:** cardiac function, childhood, acute lymphoblastic leukemia, vascular toxicity

## Abstract

Childhood acute lymphoblastic leukemia (ALL) survivors who underwent chemotherapy with anthracyclines have an increased cardiovascular risk. The aim of the study was to evaluate left and right cardiac chamber performances and vascular endothelial function in childhood ALL survivors. Fifty-four ALL survivors and 37 healthy controls were enrolled. All patients underwent auxological evaluation, blood pressure measurements, biochemical parameters of endothelial dysfunction, flow-mediated dilatation (FMD) of the brachial artery, mean common carotid intima-media thickness (c-IMT), antero-posterior diameter of the infra-renal abdominal aorta (APAO), and echocardiographic assessment. The ALL subjects had significantly lower FMD (*p* = 0.0041), higher left (*p* = 0.0057) and right (*p* = 0.0021) echocardiographic/Doppler Tei index (the non-invasive index for combined systolic and diastolic ventricular function) as compared to controls. Tricuspid annular plane excursion (TAPSE) was 16.9 ± 1.2 mm vs. 24.5 ± 3.7 mm, *p* < 0.0001. Cumulative anthracycline doses were related to TAPSE (*p* < 0.001). The ALL survivors treated with anthracyclines demonstrated systo/diastolic alterations of the right ventricle and reduced endothelial function compared with healthy controls. The early recognition of subclinical cardiac and vascular impairment during follow up is of utmost importance for the cardiologist to implement strategies preventing overt cardiovascular disease considering the growing number of young adults cured after childhood ALL.

## 1. Introduction

Pharmacological approaches to childhood acute lymphoblastic leukemia (ALL) have improved significantly in recent decades, so that a fatal disease has become one with 5-year survival rates of over 90% [1]. Therefore, the majority of ALL children are expected to become long-term survivors, although, unfortunately, many of them are still likely to experience adverse late effects related to chemotherapies [2,3,4,5,6,7]. The high incidence of cardiotoxicity is well-documented in long-term childhood ALL survivors who underwent anthracycline treatment, especially doxorubicin and daunorubicin [8,9,10]. Anthracyclines promote myocardial injury by inducing myocyte morphology and functional alterations, myocardial sarcoplasmic reticulum damage, and mitochondria vacuolization which can negatively impact on left ventricular (LV) function and compliance [11,12,13]. Depending on cumulative anthracycline doses, up to 65% of childhood ALL survivors can experience subclinical abnormalities of the LV in adulthood [14,15].

There are not many data available about the influence of chemotherapies on right ventricle (RV) function in long-term childhood ALL survivors [16,17,18]. Furthermore, although in the wide scenario of vascular endothelial damage of chemotherapy [3], some data pointed out the impact of anthracyclines on vascular function, without conclusive results [19,20,21,22].

The aim of our study was to investigate the cardiovascular function in childhood ALL survivors treated with anthracyclines, focusing on the functional parameters of subclinical damage. The early detection of chemotherapy-induced alterations both on the cardiac chambers and on the vascular walls is crucial for a better and early stratification of patients’ overall cardiovascular risk, to effectively predict cardiovascular events.

## 2. Methods

### 2.1. Study Population

Fifty-four childhood ALL survivors (35 females), mean age: 9.7 ± 4.2 (range 4–19) years, were recruited at the Paediatric Haematology and Oncology Clinic, University Hospital of Bari, Italy. Patients received treatment according to the ongoing international ALL protocols adopted by the AIEOP (Italian Association of Paediatric Haematology and Oncology). Six patients received ALL treatment according to high-risk protocols, 30 received treatment according to intermediate risk protocols, and 18 received treatment according to standard risk protocols. Nine patients were treated with cranial irradiation for central nervous system prophylaxis. Chemotherapy had been completed since at least 3 months.

Inclusion criteria were: (a) age ranging between 4–20 years; (b) ALL in complete remission; (c) end of antineoplastic therapy since at least three months. Exclusion criteria were: (a) evidence of ALL; (b) cardiovascular diseases and/or endocrine and/or metabolic disorders; (c) genetic syndromes.

As a control group, we enrolled 37 healthy subjects matched by age and sex who underwent biochemical evaluation, cardiological and vascular ultrasound (US) screening. The protocol was approved by the Ethics Committee of Policlinico of Bari (Project identification code 704, approval date: 2 July). Written informed consent was signed by all parents or patients above 18 years old. The study was conducted in accordance with the Declaration of Helsinki on Human Experimentation.

### 2.2. Auxological Parameters

Evaluation of anthropometric variables (height, weight, and waist circumference) was performed with the patients in underwear. Body weight was determined to the nearest 0.1 kg and height was measured with a Harpenden stadiometer to the nearest 0.1 cm. Body mass index (BMI) was calculated and expressed as kg/m^2^. Arterial blood pressure was measured according to the international guidelines on the Screening and Management of High Blood Pressure in Children and Adolescents [23]. Puberty was classified according to Tanner’s staging system [24].

### 2.3. Biochemical and Hemostatic Markers

Fasting glycemia, insulin, total cholesterol (TC), high (HDL-C) and low (LDL-C) density lipoprotein cholesterol, and triglycerides (TG), were measured after overnight fasting in all subjects. Insulin resistance was assessed by the HOMA index (HOMA-IR) [25]. Commercial ELISA tests were used for total adiponectin and multimeric high-molecular weight (HMW) subfraction (ELISA 47-ADPH-9755; ALPCO Diagnostics, Salem, VT, USA), and Endothelin-1 levels (R&D System Europe, Lille, France).

### 2.4. Cardiovascular US Measurement

Cardiovascular US studies included measurement of mean common carotid intima-media thickness (mean-IMT), flow mediated dilatation (FMD) of the brachial artery, and antero-posterior abdominal aorta diameter (APAO), while M-Mode, B-Mode Echocardiography, and Tissue Doppler Imaging (TDI) were performed in order to investigate cardiac morphology and function, as previously described [26].

### 2.5. Evaluation of Mean-IMT

Ultrasonographic echo-color Doppler studies of left and right common carotid arteries were performed bilaterally by the same physician with a Philips Sonos 5500 using a 7.5 MHz high resolution probe. The patients were placed in a supine position, with the neck extended and rotated contra-laterally by 45°: common carotid arteries were examined on the sagittal axis in lateral view. Mean-IMT was defined as the low-level echo gray band that does not project into the arterial lumen and was measured during end-diastole according to the method described by Pignoli et al. [27]. The measurements were bilaterally performed 1 cm proximally to the carotid bulb, three times each, and then mean-IMT was calculated. Agreement between measurements was very high, 0.98 according to the intra-class correlation coefficient (ICC; good if >0.80) [28].

### 2.6. Evaluation of FMD

The FMD of the brachial artery was non-invasively assessed using a high-resolution ultrasound probe in a quiet, air-conditioned environment (22–24 °C). The subjects were fasted for at least 8–12 h. A dedicated software, certified by the CNR of Pisa (MVE II), analyzed the images. Measurements were made by the same observer to reduce biases. The subjects were studied in the supine position. A sphygmomanometer cuff was inserted close to the brachial artery. After 1 min of basal image acquisition, the cuff was inflated till 200–220 mmHg to promote an ischemic stimulus. After 5 min, we deflated the cuff: the increase in shear stress results in production of nitric oxide (NO) which provides the stimulus for vasodilation. After 15 s from the end of ischemia, the flow rate was measured and then the degree of hyperemia. The FMD was calculated as the ratio between the changes in diameters of the brachial artery (maximum expansion after deflation baseline/baseline diameter) divided by its basal value and it was expressed as percentage [29]. The measure of the FMD showed good reproducibility in our study, with a calculated ICC of about 0.95. Antero-posterior diameter of infrarenal abdominal aorta (APAO) was performed by a single operator using a single high-resolution vascular ultrasound Philips 5500 equipped with a 3 MHz electronic probe. The electronic probe was placed one centimeter left of the umbilicus. Then, the best image in long-axis projection of the abdominal aorta was obtained. The APAO was defined as the maximal external cross-sectional diameter measurement of the infrarenal abdominal aorta. It was calculated as the distance between the near and the far walls of the abdominal aorta [30,31]. APAO showed a good reproducibility in our study as we found an ICC of about 0.92.

### 2.7. Echocardiography and TDI

All patients underwent echocardiography of both left and right chambers, in agreement with international guidelines [32,33]. Pulsed wave TDI was used to evaluate the velocity of the ventricle walls and the related parameters of systolic and diastolic function of both LV and RV [29]. The cardiac structures examined were mitral valve annulus, basal and mid part of the LV lateral wall, interventricular septum, basal part of the RV lateral wall, and the lateral tricuspid annulus [34]. We calculated mitral and tricuspid E/A ratio and E/E’ ratio. Systolic and diastolic time parameters related to both RV and LV were measured throughout the entire cardiac cycle [35,36]. Left (l-IVCT) and right (r-IVCT) isovolumetric contraction time, left (l-ET) and right (r-ET) ejection time, and left (l-IVRT) and right (r-IVRT) isovolumetric relaxation time were measured to obtain the LV and RV TEI index ((IVRT + IVCT)/ET) [28]. The calculated ICC for all the echocardiographic measurements was 0.87.

### 2.8. Statistical Analysis

We applied the *t*-test to compare the mean values of variables between cases and controls, with the Welch modification to account for heteroscedasticity. Linear regression models were exploited to assess the association between dose and cardiovascular outcomes in the cases. Each regression model included the other variables as confounders and the lasso penalty was applied to the discarded nonsignificant ones’ heteroscedasticity [37]. Multivariate analysis with linear regression model was used to determine the effect of cumulative anthracycline doses and vascular and cardiac parameters. The limit of statistical significance was set at 0.05.

## 3. Results

Baseline and cardiovascular characteristics of the study population are listed in Table 1. No significant differences were found between ALL children and controls according to age. Systolic blood pressure (SBP) and diastolic blood pressure (DBP) were not different between the two groups. Indeed, ALL patients showed higher BMI and waist circumference than controls. Biochemical measurements also exhibited higher fasting glucose levels in ALL patients than controls, although no differences were found according to insulin and HOMA-IR measurements. Furthermore, total cholesterol, LDL-cholesterol, HDL-cholesterol, and triglycerides were statistically different between patients and controls. In addition, the ALL patients showed a statistically significant reduction in adiponectin and HMW adiponectin than controls, while no difference emerged in the endothelin-1 levels.

Table 2 shows the vascular and cardiac findings in ALL patients and controls. Brachial artery FMD measurements were significantly lower in ALL patients, while no differences were found according to morphological alterations in vascular walls as the values of mean IMT and APAO were similar between the two groups.

According to cardiac parameters, patients and controls did not show a significant difference in left ventricle ejection fraction (LVEF). Nevertheless, both left and right TEI index measurements were higher in ALL patients than controls (0.42 ± 0.07 vs. 0.36 ± 0.11, *p* = 0.0057; 0.43 ± 0.10 vs. 0.34 ± 0.14, *p* = 0.0021, respectively). Tricuspid annular plane excursion (TAPSE) measurements were lower in ALL patients than controls (16.9 ± 1.2 mm vs. 24.5 ± 3.7 mm) both in the univariate (*p* < 0.0001) and multivariate analysis (*p* <0.0001, linear regression model). Figure 1 portrays the box plots comparing the distributions of the cardiovascular and vascular variables having significantly different mean values between ALL patients and controls.

With regards to the cumulative doses of anthracyclines, they were 210 mg/mq in 1/54 patient, 240 mg/mq in 48/54 patients (assigned to the standard and intermediate risk protocols), 310 mg/mq in 1/54 patient, and 350 mg/mq in 4/54 patients, assigned to the high-risk protocol. In the multivariate analysis (linear regression model), the cumulative anthracycline dose (mg/m^2^) results related to TAPSE: the higher the cumulative dose, the worst the systolic function of the right ventricle, as assessed by the inverse relationship between the two variables (*p* < 0.001) (Figure 2).

## 4. Discussion

This study demonstrates the subclinical impairment of both left and right cardiac and vascular function in childhood ALL survivors treated with anthracycline-based regimens. The assessment of RV function recently gained increasing interest within cardio-oncology due to its role in predicting the occurrence of heart failure [38,39,40,41,42,43]. In our sample of ALL childhood survivors, subclinical alterations to the RV function were outlined by the higher values of right TEI index and lower TAPSE measurements. Interestingly, the higher cumulative doses of anthracyclines corresponded with significantly lower values of TAPSE. To our knowledge, such an intriguing result has not been reported before. Christiansen et al. [16] found reduced values in TAPSE in adult survivors of childhood malignant lymphoma or ALL who had been exposed to anthracyclines, mediastinal radiotherapy, or both. In particular, a global reduction in RV function, as outlined by alterations also in fractional area change, peak systolic tricuspid annular velocity, and free wall strain, was observed. In line with our results, Christiansen et al. found no variations in terms of right diastolic function between ALL patients and controls [16]. Cardiotoxicity has been associated mainly to anthracycline doses >300 mg/mq [36]. However, signs of cardiotoxicity have also been reported with lower anthracycline doses of 100 mg/mq [14,44,45,46]. In the present study, anthracycline doses varied from 210 mg/mq to 350 mg/mq, therefore some cardiotoxicity might be expected in our sample. Bayram et al. [44] observed reduced right ventricle myocardial velocities, as assessed by TDI, in childhood leukemia survivors treated with low doses of anthracyclines, thus deriving the impairment in systolic and diastolic function of the cardiac chamber, but do not provide any data about the right TEI index. Interestingly, Kocabaş et al. [47] outlined the progressive increase in the right TEI index with the increase in cumulative anthracycline doses. Although our study did not demonstrate the correlation between right TEI index increase and cumulative dose, the identification of a progressive decrease in TAPSE with the increase in cumulative anthracycline doses confirmed the impairment in the right ventricle function in ALL survivors. Indeed, patients treated with anthracyclines were reported to show impairment in right ventricular systolic and diastolic functional reserves when they underwent stress echocardiography; thus, they had subtle alterations in the right cardiac chamber which can be exacerbated by stress [48,49,50,51]. All these findings should be taken into account as the RV function seems to be a stronger predictor of developing or worsening heart failure than the LV function [52].

Cardiovascular toxicity due to anticancer therapies may range from asymptomatic and transient to clinically significant and may be long-lasting. Guidelines have been drawn up to help clinicians in the prevention and delicate management of cardiovascular toxicity [53,54]

Previous studies have focused on the remodeling of left cardiac chambers during chemotherapy [11,12,13,14,36].

A worsening of LV function was observed in 7% of ALL patients after a mean period of 6 years after chemotherapy [55]. Conversely, in our study, we did not observe statistically significant differences in terms of LVEF between ALL patients and controls. However, ALL patients showed higher values in left TEI index than controls, which can be considered an early sign of systo-diastolic alteration in left cardiac chambers. In particular, left TEI index values >0.38 may distinguish ALL survivors from healthy controls. This finding agrees with the previous literature data [19,44,47,50,51,52,55,56].

We did not demonstrate alterations in vascular morphology in ALL survivors as compared to controls. Mean IMT and APAO were similar between the two groups. ALL survivors showed a statistically significant decrease in endothelial function as expressed by FMD. Derangement in endothelial function after anthracycline treatment has been previously reported in cancer survivors [3,57,58,59,60,61,62,63]. Jenei et al. [19] outlined the occurrence of both endothelial dysfunction and increased aortic stiffness in long-term survivors of childhood cancer, both related to cumulative anthracycline dose (in mg/m^2^). Long et al. [61] confirmed the reduction in FMD values and subclinical left ventricle diastolic dysfunction during exercise stress in patients treated with anthracyclines. In our study, we did not find any correlation of vascular parameters with different cumulative doses of anthracyclines.

Finally, as regards the metabolic markers of endothelial dysfunction, namely HMW-AD, endothelin-1, and insulin resistance [64,65], in our study population we confirmed the reported data on impaired metabolic profile in young survivors of ALL [3]; however, we did not find any correlations with cardiovascular parameters and total doses of anthracyclines.

Childhood ALL survivors treated with anthracyclines showed contemporary early echocardiographic signs of RV and endothelial dysfunctions. Detection of early cardiac and vascular impairment should be mandatory to prevent progression of the cardiovascular toxicity and thus improving the long-term quality of life of this growing population of young adults.

## Figures and Tables

**Figure 1 children-09-01731-f001:**
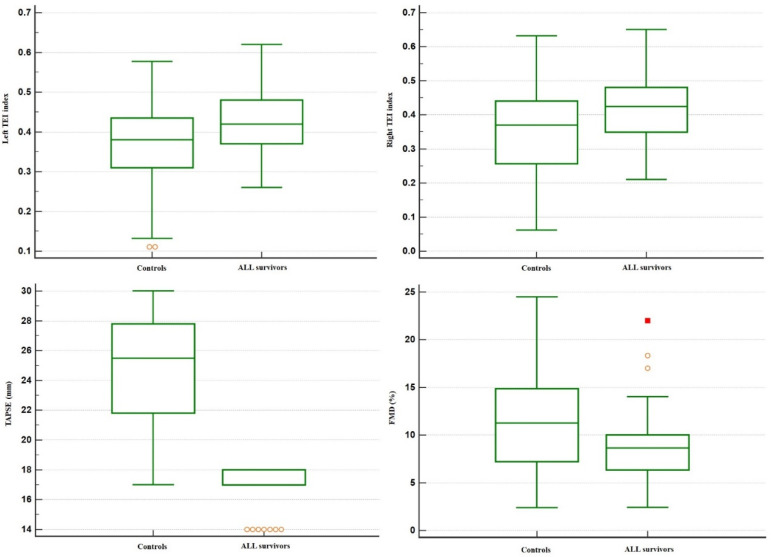
Box plots of left and right TEI index, tricuspid annular plane excursion (TAPSE), and flow-mediated vasodilatation of the brachial artery (FMD) in healthy controls and childhood cancer survivors. The orange circles and red square represent the outliers.

**Figure 2 children-09-01731-f002:**
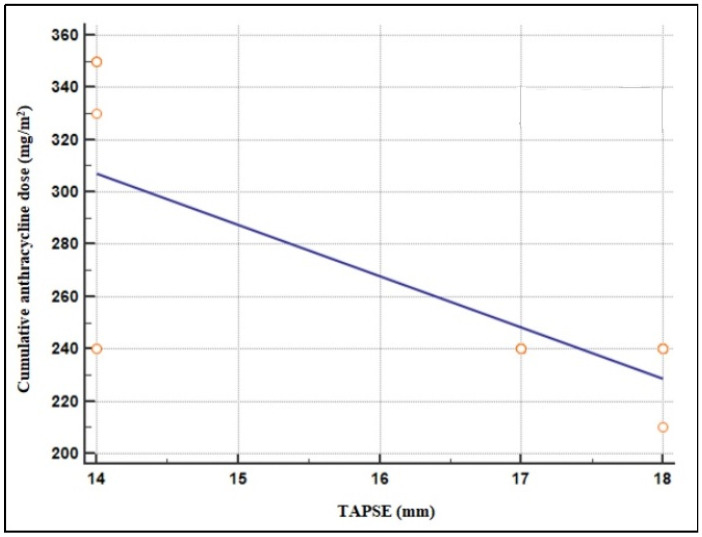
Inverse relationship between cumulative anthracycline dose (mg/m^2^) and the systolic function of the right ventricle.

**Table 1 children-09-01731-t001:** Anthropometric, clinical, and laboratory characteristics of the study population and controls.

Characteristic	ALL Patients (*n* = 54)	Controls (*n* = 37)	*p* Value
Age (years)	9.9 ± 4.2	10.3 ± 2.8	0.52
BMI (Kg/m^2^)	21.1 ± 4.8	17.5 ± 1.9	<0.0001
Waist circumference (cm)	71.5 ± 15.3	55.1 ± 7.9	<0.0001 *
SBP (mmHg)	105.6 ± 10.3	104.8 ± 5.7	0.63 *
DBP (mmHg)	66.0 ± 7.6	63.8 ± 5.7	0.11
Fasting glucose level (mg/dL)	82.3 ± 6.4	77.4 ± 6.2	<0.0001 *
Insulin (uIU/mL)	12.5 ± 8.6	13.0 ± 2.6	0.70
HOMA-IR	2.6 ± 2.0	2.5 ± 0.6	0.79
Total Cholesterol (mg/dL)	150.7 ± 23.7	126.4 ± 13.0	<0.0001
LDL-C (mg/dL)	86.4 ± 18.0	59.8 ± 14.1	<0.0001
HDL-C (mg/dL)	50.6 ± 9.9	58.9 ± 8.2	<0.0001 *
Triglycerides (mg/dL)	68.3 ± 30.6	48.2 ± 15.3	<0.0001
Endothelin-1 (pg/mL)	2.1 ± 0.6	2.0 ± 0.5	0.69
Adiponectin (μg/mL)	7.8 ± 3.0	9.0 ± 2.0	0.0288
HMW Adiponectin (μg/mL)	4.4 ± 2.4	5.8 ± 1.8	0.0059

Data are expressed as mean ± standard deviation. * = statistically significant association confirmed at multivariate analysis, i.e., after adjusting for other variables.

**Table 2 children-09-01731-t002:** Echocardiographic and vascular characteristics of the study population and controls.

Characteristic	ALL Patients (*n* = 54)	Controls (*n* = 37)	*p* Value
FMD (%)	8.7 ± 3.5	11.6 ± 5.0	0.0041
Mean IMT (mm)	0.46 ± 0.07	0.45 ± 0.06	0.22
APAO (mm)	10.1 ± 2.0	10.2 ± 1.9	0.71
LVEF (%)	59.7 ± 8.6	61.7 ± 6.3	0.22
Left TEI index	0.42 ± 0.07	0.36 ± 0.11	0.0057
Right TEI index	0.43 ± 0.10	0.34 ± 0.14	0.0021
Left E/A ratio	2.2 ± 0.7	2.0 ± 0.5	0.15
Right E/A ratio	1.9 ± 0.6	1.9 ± 0.6	0.97
TAPSE (mm)	16.9 ± 1.2	24.5 ± 3.7	<0.0001 *

Data are expressed as mean ± standard deviation. * = statistically significant association confirmed at multivariate analysis, i.e., after adjusting for other variables.

## Data Availability

Not applicable.

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
