# Peer review of "Assessment of Cardiovascular Function in Childhood Leukemia Survivors: The Role of the Right Heart"

_children, 2022, doi:10.3390/children9111731_

Round 1

Reviewer 1 Report

Dear authors,

In general it is a well written manuscript about an important topic of research in Cardiology. Congratulations for your novel and interesting work. My comments for you are the following:

1. You could include in the section of discussion the guidelines about diagnosis/management of patients who have received chemotherapy and have developed cardiac damage.

2. You should improve the legends of tables (please, explain in detail every abbreviation).

3. Please, correct some minor typographic errors (i.e. line 228).

4. Please, improve the reference style according to the journals needs.

Author Response

Review Report 1

Dear authors,

In general, it is a well written manuscript about an important topic of research in Cardiology. Congratulations for your novel and interesting work. My comments for you are the following:

  1. You could include in the section of discussion the guidelines about diagnosis/management of patients who have received chemotherapy and have developed cardiac damage.

Dear reviewer, we really appreciate your comments and we would like to thank you.  A sentence about recent ESC guidelines on cardio-oncology has been added in the discussion with the corresponding reference.

  1. You should improve the legends of tables (please, explain in detail every abbreviation).

The table legends have been detailed.

  1. Please, correct some minor typographic errors (i.e. line 228).

Minor errors have been corrected.

  1. Please, improve the reference style according to the journal needs.

The reference style has been entirely revised and corrected.

Reviewer 2

Dear reviewer, thank you very much for your comments, corrections have been made as follows:

line 25, uncertain TEI index definition

A definition of Tei index has been added, moreover a reference from Tei C. has been added in the methods

line 212, exacerbated

Correction made

I appreciate the novel finding of worse RV systolic function compared to LV function and the presence of endothelial dysfunction as a measure of poorer overall impaired vascular health

table II no definition of TEI index in labels

An explanation of Tei index has been added in the table legend

Reviewer 2 Report

line 25, uncertain TEI index definition

line 212, exacerbated

I appreciate the novel finding of worse RV systolic function compared to LV function and the presence of endothelial dysfunction as a measure of poorer overall impaired vascular health

table II no definition of TEI index in labels

Author Response

Reviewer 2

Dear reviewer, thank you very much for your comments, corrections have been made as follows:

line 25, uncertain TEI index definition

A definition of Tei index has been added, moreover a reference from Tei C. has been added in the methods.

line 212, exacerbated

Correction made.

I appreciate the novel finding of worse RV systolic function compared to LV function and the presence of endothelial dysfunction as a measure of poorer overall impaired vascular health

table II no definition of TEI index in labels

An explanation of Tei index has been added in the table legend.
